Identification and experimental verification of necroptosis-related prognostic gene signature and characterization of tumor immune infiltration in lung squamous cell carcinoma

Sun Kai 1
Wang Ke-run 1
Wen Song 1
Hong Juan-juan 2
Fei Yu-lang 3
Pan Qing-hua panqingh2025@yeah.net 1
Xie Fang-fang xffgzsrmyy@yeah.net 4
1 The Affiliated Cancer Hospital of Gannan Medical University (Ganzhou Cancer Hospital) , Ganzhou , China
2 Liuzhou People’s Hospital , Liuzhou , Guangxi Zhuang Autonomous Region , China
3 The First Affiliated Hospital of Nanyang Medical College , Nanyang , Henan Province , China
4 Department of Rheumatology, Ganzhou People’s Hospital , Ganzhou , Jiangxi Province , China
Khan Imran
Electronic publication date: 2025 Oct 29
Publication date: 2025
Volume: 13
Electronic Location ID: e20260
Received 2025 May 29; Accepted 2025 Sep 29
Copyright: ©2025 Sun et al.
Copyright year: 2025
Copyright holder: Sun et al.
License: This is an open access article distributed under the terms of the Creative Commons Attribution License, which permits unrestricted use, distribution, reproduction and adaptation in any medium and for any purpose provided that it is properly attributed. For attribution, the original author(s), title, publication source (PeerJ) and either DOI or URL of the article must be cited.
License URL: https://creativecommons.org/licenses/by/4.0/

Keywords: Lung squamous cell carcinoma, Necroptosis, Prognosis signature, Immunohistochemistry, Tumor immune infiltration

Funding: The Science and Technology Development Planning Project of Ganzhou GZWJW202502271 The Science and Technology Development Planning Project of Nanyang 23JCQY2055 This work was supported by the Science and Technology Development Planning Project of Ganzhou (GZWJW202502271) and the Science and Technology Development Planning Project of Nanyang (23JCQY2055). There was no additional external funding received for this study. The funders had no role in study design, data collection and analysis, decision to publish, or preparation of the manuscript.

==============================
Background

Lung squamous cell carcinoma (LUSC) is a common and aggressive malignancy. Necroptosis, a regulated mode of cell death, has been implicated in tumor immunity and oncogenic processes, yet the mechanistic involvement of necroptosis-related genes (NRGs) in LUSC pathogenesis remains unclear, necessitating systematic evaluation of their biological and clinical relevance.

Methods and Results

Clinical and transcriptomic data of LUSC patients from The Cancer Genome Atlas (TCGA) and Gene Expression Omnibus (GEO) were subjected to integrative analyses. Screening of the Kyoto Encyclopedia of Genes and Genomes (KEGG) database identified 159 NRGs, among which 35 differentially expressed NRGs (DENRGs) were associated with necroptosis, apoptosis, and immune signaling pathways. Cox regression combined with Least Absolute Shrinkage and Selection Operator (LASSO) analysis yielded three NRGs (CAMK2A, CHMP4C, and PYGB) strongly associated with patient prognosis. Based on these genes, a prognostic model was constructed to stratify patients into high- and low-risk subgroups with distinct survival patterns. External dataset validation demonstrated moderate predictive accuracy. Reverse transcription-quantitative polymerase chain reaction (RT-qPCR) and immunohistochemistry (IHC) confirmed abnormal expression of the three genes in LUSC tissues. Additional analyses revealed correlations of these NRGs with immune infiltration, immune checkpoint activity, tumor mutation burden (TMB), and microsatellite instability (MSI).

Conclusions

A three-gene NRG signature was identified as a prognostic marker in LUSC. These genes appear to influence disease progression and the immune microenvironment, highlighting their potential as therapeutic targets and as a foundation for further investigation.

Introduction

Lung cancer remains the leading cause of cancer mortality in the USA, with an estimated 124,730 deaths projected for 2025, representing 20% of all cancer-related fatalities. The anticipated incidence for the same year is 226,650 cases. In 2021, women under 65 exhibited a higher incidence than men (15.7 vs. 15.4 per 100,000), marking the first female predominance in this cohort since the pre–tobacco era (Siegel et al., 2025). Non-small-cell lung cancer (NSCLC) accounts for the vast majority of lung cancers, with lung squamous cell carcinoma (LUSC) ranking as the second most common subtype after adenocarcinoma (Shen, Chen & Li, 2025). The difficulty of achieving timely diagnosis in LUSC contributes to its persistently poor 5-year survival rate (Niu et al., 2022b). Standard therapeutic approaches include surgery, chemoradiotherapy, and targeted therapy (Chen et al., 2024). More recently, immunotherapy has significantly reshaped the treatment landscape of LUSC, as anti-PD-L1 agents have demonstrated durable survival advantages in selected patients (Niu et al., 2022b). In recent years, immunotherapy has markedly advanced the treatment landscape of LUSC, with anti-PD-L1 agents demonstrating durable survival benefits in selected patient populations. However, the discovery of reliable biomarkers for predicting therapeutic efficacy remains controversial and continues to be limited in LUSC research. Current evidence indicates that co-wild-type (co-WY) TP53 and LRP1B are correlated with higher tumor mutation burden (TMB), and such genomic profiles are associated with superior survival outcomes in patients receiving anti-PD-L1 therapy (Yu et al., 2022). These findings suggest the potential of co-WY TP53 and LRP1B as biomarkers for optimizing immunotherapeutic decision-making in LUSC. In parallel, the microsatellite instability-associated prognostic risk score (MSI-pRS) has been introduced as a candidate prognostic marker. A low MSI-pRS has been linked to elevated genomic instability and an immunologically cold phenotype, providing an additional perspective for biomarker exploration in LUSC (Hu et al., 2023). Beyond biomarker development, ultrasound-mediated nanobubble technology has enabled the targeted delivery of STAT6 siRNA to M2 tumor-associated macrophages (M2-TAMs), effectively suppressing the TGF-β1-EMT axis in LUSC cells. This minimally invasive strategy not only inhibits M2-TAM polarization but also broadens therapeutic opportunities for LUSC management (Shu et al., 2024). Collectively, although therapeutic challenges persist, progress is being driven by investigations into molecular mechanisms and the exploration of innovative treatment strategies. Future efforts should emphasize biomarker validation and the development of novel modalities to improve survival and quality of life in patients with LUSC.

Apoptosis remains the principal mechanism through which most anticancer drugs exert their effects, and suppression of this process in cancer cells confers resistance to multiple chemotherapeutic agents, thereby reducing therapeutic efficacy (Gielecińska et al., 2023). Continuous investigation into cell death pathways has revealed an expanding spectrum of regulatory mechanisms (Guo et al., 2025). Among them, necroptosis represents a programmed form of necrosis characterized by cell swelling, plasma membrane rupture, and organelle disintegration, sharing morphological traits with both necrosis and apoptosis (Yuan, Amin & Ofengeim, 2019). Its regulation depends on receptor-interacting protein kinase 1 (RIPK1), receptor-interacting protein kinase 3 (RIPK3), and mixed-lineage kinase domain-like protein (MLKL) (Zhang et al., 2022). Tumor necrosis factor (TNF-α) initiates necroptosis by binding to cell surface receptors, which induces phosphorylation and assembly of RIPK1 and RIPK3 into necrosomes (Yuan, Amin & Ofengeim, 2019; Zhang et al., 2022). Subsequent MLKL activation disrupts membrane integrity, releasing intracellular contents into the extracellular space. The RIPK1/RIPK3/MLKL signaling cascade constitutes a central axis governing necroptotic cell death in tumor biology (Martens et al., 2021).

Necroptosis has gained increasing attention for its involvement in diverse pathological conditions, including malignancies. In the context of LUSC, its therapeutic relevance derives from the ability to induce cell death independently of caspase activity, a pathway often disrupted in cancer cells (Zhang et al., 2022; Meier et al., 2024). The influence of necroptosis in cancer, particularly LUSC, is complex and context-dependent, functioning as a double-edged mechanism (Dhuriya & Sharma, 2018; Ye, Chen & Xu, 2023). While induction of necroptosis can suppress tumor growth through direct elimination of cancer cells, its pro-inflammatory characteristics may simultaneously promote tumor progression and metastasis by sustaining a tumor-supportive microenvironment (Ye, Chen & Xu, 2023). Understanding the molecular determinants that govern necroptosis in LUSC therefore represents a prerequisite for therapeutic exploitation. Emerging evidence highlights the therapeutic promise of manipulating necroptotic signaling. For instance, stabilization of RIPK3 through SPOP-mediated ubiquitination enhances necroptotic cell death in colon cancer, suggesting that comparable processes may operate in LUSC (Lee et al., 2024). Moreover, crosstalk between necroptosis and other regulated cell death modalities, including ferroptosis and pyroptosis, expands the therapeutic landscape. Activation of immunogenic cell death through these pathways holds potential to enhance the efficacy of immune checkpoint inhibitors, whose clinical benefit in LUSC remains limited (Niu et al., 2022a). Additional regulation arises from cellular proteins such as Bcl-2, which suppresses MLKL oligomerization and thereby interferes with necroptotic initiation (Shi & Kehrl, 2019). Elucidation of these regulatory layers is essential to designing therapeutic strategies that selectively induce necroptosis in cancer cells while mitigating deleterious inflammatory consequences.

Necroptosis has been recognized as a determinant of the TIME, influencing both tumor progression and therapeutic responsiveness (Meier et al., 2024; Zang et al., 2022; Zhang et al., 2024). Evidence indicates that necroptosis induces the release of damage-associated molecular patterns (DAMPs), which activate immune signaling and promote infiltration of immune cells into tumors (Luo et al., 2023). This mechanism contributes to the recruitment of cytotoxic T cells and other effector populations, thereby enhancing anti-tumor immunity and improving therapeutic efficacy. Moreover, necroptosis regulates immune checkpoint pathways such as PD-1 and CTLA-4, thereby shaping the effectiveness of immune checkpoint blockade (Meier et al., 2024; Yu et al., 2024). Consequently, necroptosis emerges as a therapeutically relevant yet complex target in LUSC, where its dual tumor-restraining and tumor-promoting functions necessitate context-specific strategies. Clarifying its conditional roles in LUSC and devising approaches to precisely modulate this pathway remain essential for advancing therapeutic outcomes.

The present study was designed to investigate the contribution of necroptosis to LUSC progression and its interplay with the tumor immune microenvironment, addressing unresolved gaps in current knowledge. The analysis aimed to improve the identification of diagnostic and prognostic biomarkers for LUSC by examining necroptosis-associated mechanisms. To this end, the study established a necroptosis-related prognostic signature based on The Cancer Genome Atlas (TCGA) and Gene Expression Omnibus (GEO) datasets. Expression profiles of selected necroptosis- related genes (NRGs) were further evaluated in LUSC and normal lung tissues through reverse transcription-quantitative polymerase chain reaction (RT-qPCR) and immunohistochemistry (IHC). In addition, the relationship between NRGs and tumor microenvironment infiltration was systematically assessed. Collectively, the results provide a potential framework for biomarker discovery and prognostic prediction in LUSC.

Materials and Methods

Datasets and data processing

This investigation employed a retrospective design, with experimental data derived from methodologies previously established (Sun et al., 2022). Transcriptomic profiles (RNA-Seq FPKM values), demographic variables, and survival outcomes of TCGA-LUSC cases were retrieved from the UCSC Xena repository (https://xena.ucsc.edu/; complete metadata) (Goldman et al., 2020). Clinical and pathological parameters associated with NRG expression in LUSC were systematically summarized in Table 1. To strengthen analytical reliability, additional LUSC cohorts were incorporated from the Gene Expression Omnibus (GEO, https://www.ncbi.nlm.nih.gov/geo/), including GSE19188 (24 LUSC samples), GSE41271 (80 LUSC samples), GSE42127 (43 LUSC samples), GSE81089 (67 LUSC samples), GSE157009 (249 LUSC samples), GSE30219 (61 LUSC samples), GSE73403 (69 LUSC samples), GSE8894 (75 LUSC samples), and GSE74777 (107 LUSC samples) (Barrett et al., 2013). Independent validation cohort with comparable clinicopathological characteristics. These datasets were selected because they provide comprehensive gene expression data and clinical information, making them suitable for external validation. A curated set of 159 NRGs was extracted from the KEGG database, with complete expression profiles provided in Table S1 (Kanehisa & Goto, 2000). Validation was conducted through cross-referencing with published literature and empirical evidence concerning necroptotic signaling pathways. All computational procedures were implemented in R statistical software (version 4.0.3).

Table 1 Clinical characteristics of patients with LUSC in TCGA databases.

Characteristic	levels	Overall	
n		501	
T stage, n (%)	T1	114 (22.8%)	
	T2	293 (58.5%)	
	T3	71 (14.2%)	
	T4	23 (4.6%)	
N stage, n (%)	N0	319 (64.4%)	
	N1	131 (26.5%)	
	N2	40 (8.1%)	
	N3	5 (1%)	
M stage, n (%)	M0	411 (98.3%)	
	M1	7 (1.7%)	
Pathologic stage, n (%)	Stage I	244 (49.1%)	
	Stage II	162 (32.6%)	
	Stage III	84 (16.9%)	
	Stage IV	7 (1.4%)	
Primary therapy outcome, n (%)	PD	31 (8.6%)	
	SD	17 (4.7%)	
	PR	5 (1.4%)	
	CR	308 (85.3%)	
Gender, n (%)	Female	130 (25.9%)	
	Male	371 (74.1%)	
Race, n (%)	Asian	9 (2.3%)	
	Black or African American	30 (7.7%)	
	White	349 (89.9%)	
Age, n (%)	<=65	190 (38.6%)	
	>65	302 (61.4%)	
number_pack_years_smoked, n (%)	<40	133 (31.4%)	
	>=40	291 (68.6%)	
Smoker, n (%)	No	18 (3.7%)	
	Yes	471 (96.3%)	
OS event, n (%)	Alive	285 (56.9%)	
	Dead	216 (43.1%)	
DSS event, n (%)	Alive	360 (80.2%)	
	Dead	89 (19.8%)	
PFI event, n (%)	Alive	354 (70.7%)	
	Dead	147 (29.3%)	
Age, median (IQR)		68 (62, 73)	

NRGs differential expression, survival and mutation analysis

Comprehensive bioinformatics evaluation identified 35 differentially expressed NRGs (DENRGs) with significant differential expression in LUSC (Table S2). Expression profiles of tumor and adjacent normal lung tissues were compared using the limma package with stringent cutoffs (—log2FC—>1, adjusted P-value < 0.05) (Liu et al., 2021). Visualization of results employed several R packages: ComplexHeatmap for hierarchical clustering, Enhanced Volcano for differential expression mapping, ggplot2 for quantitative profiling, and survminer for Kaplan–Meier curve construction (Ito & Murphy, 2013; Gu, 2022). Survival outcomes were further assessed using the log-rank test in conjunction with Cox proportional hazards regression. In addition, mutational features of NRGs in LUSC were systematically characterized with the maftools package (Mayakonda et al., 2018).

Functional annotation and pathway analysis

To clarify the biological relevance of differentially expressed NRGs, functional enrichment analyses were conducted. Gene Ontology (GO) annotation delineated major biological processes, molecular functions (MF), and cellular components (CC) linked to these genes (The Gene Ontology Consortium, 2019). KEGG pathway analysis further mapped the associated systemic functional networks. Both analyses were graphically represented and interpreted using ggplot2 to provide an integrated view of enrichment outcomes.

Development and validation of necroptosis-related prognostic signature

Candidate NRGs with prognostic significance were identified through univariate and multivariate Cox regression analyses, yielding independent biomarkers for model construction. Genes selected at λ.1se were entered into a multivariate Cox proportional hazards framework, and only statistically significant variables (P < 0.05) were preserved in the final model. A Least Absolute Shrinkage and Selection Operator (LASSO) regression approach incorporating three genes was applied to establish a prognostic signature. Risk scores were calculated as NRGs_score = Σ(Expi × coefi), and patients were classified into high- or low-risk groups based on the median cutoff. Kaplan–Meier survival analysis and ROC curve evaluation were employed to assess predictive accuracy (Kamarudin, Cox & Kolamunnage-Dona, 2017). A nomogram integrating clinical factors was further generated to predict 1-, 3-, and 5-year survival. All analyses were conducted using dedicated R packages, and independent GEO cohorts served as external validation datasets.

Cell lines and cell culture

The BEAS-2B human bronchial epithelial cell line and the NCI-H520 human LUSC cell line were obtained from Sangon Biotech (Shanghai, China). BEAS-2B cells were cultured in Dulbecco’s Modified Eagle Medium (DMEM; Gibco) containing 10% fetal bovine serum (FBS; Gibco), whereas NCI-H520 cells were maintained in Roswell Park Memorial Institute 1640 medium (RPMI-1640; Gibco) supplemented with 10% FBS (Gibco). All cultures were incubated at 37 °C in a humidified environment with 5% CO2.

RNA extraction and RT-qPCR assay

mRNA expression of CAMK2A, CHMP4C, and PYGB was assessed by RT-qPCR. Total RNA was isolated with TRIzol reagent (Ambion) following standard procedures, and complementary DNA (cDNA) was synthesized using 5 × HiScript qRT SuperMix II (Vazyme Biotech). Quantitative amplification was carried out on an Mx3000P QPCR system (Agilent Technologies, Santa Clara, CA) with ChamQ Universal SYBR qPCR Master Mix (Vazyme Biotech) under the manufacturer’s recommended conditions. The cycling protocol consisted of an initial denaturation at 95 °C for 10 s, followed by 40 cycles of 95 °C for 15 s and 60 °C for 60 s for annealing/extension. Relative expression levels were calculated using the 2−ΔΔCt method, with GAPDH as the internal reference. Each sample was analyzed in triplicate to ensure reproducibility. Primer sequences for GAPDH and the target genes were as follows: GAPDH: Sequence (5′→3′).

Forward: TCAAGAAGGTGGTGAAGCAGG, Reverse: TCAAAGGTGGAGGAGTGGGT, 115 bp.

CAMK2A: Sequence (5′->3′).

Forward: CAGAAGTGCTGCGGAAGG, Reverse: ATGCGTTTGGATGGGTTA, 235 bp.

CHMP4C: Sequence (5′->3′).

Forward: GTTGGCTTTGGTGATGACTT, Reverse CTGGTTTTCTATTTGGCTGT, 148 bp.

PYGB: Sequence (5′->3′).

Forward: ACGGCTATGGAATCCGCTAT, Reverse: TGTTGACGGTGTTGTTCTTGT, 255 bp.

Clinical specimen analysis and immunohistochemical validation

A total of 21 paired LUSC and adjacent non-tumorous tissues were collected from surgical resections at Liuzhou People’s Hospital between 2019 and 2021. This study was approved by the Medical Ethics Committee of Liuzhou People’s Hospital (Reference No. KY2021-025-02) and was performed according to the Declaration of Helsinki, and written informed consent was obtained from all participants. After routine tissue preparation, immunohistochemical staining was carried out with antibodies against CAMK2A, CHMP4C, and PYGB. Quantification of staining intensity was achieved by calculating integrated optical density (IOD) values through Image-ProPlus6.0 software. The corresponding clinical parameters are presented in Table S3.

Immune cell infiltration, immune checkpoints, TMB and MSI analysis

The R packages “GSVA,” “immunedeconv,” “estimate,” “ggplot2,” “pheatmap,” and “ggstatsplot” were applied to evaluate the relationship between the five prognostic NRGs and immune cell infiltration, using three advanced algorithms: ssGSEA, ESTIMATE, and CIBERSORT (Chen et al., 2022; Chen et al., 2018). These approaches, together with immune profiling of malignant tumor tissues, provide essential insights into the tumor microenvironment (TME) and its relevance for prognosis and therapeutic response. ssGSEA, an adaptation of the Gene Set Enrichment Analysis (GSEA) algorithm, generates enrichment scores for each sample–gene set pair rather than across grouped samples and pathways (https://www.genepattern.org/modules/docs/ssGSEAProjection/4). ESTIMATE calculates tumor purity and quantifies stromal and immune cell infiltration from expression data, with results accessible across TCGA tumor types and platforms. CIBERSORT deconvolutes bulk RNA-sequencing data using a reference gene expression matrix to estimate the relative abundance of 22 immune cell types, thereby enabling detailed characterization of immune landscapes and their association with disease features or therapeutic outcomes. In addition, stromal, immune, and ESTIMATE scores were examined in parallel with sixty immune checkpoint molecules, 150 marker genes from five immune pathways, TMB, and MSI. All statistical analyses and data visualizations were conducted using R version 4.0.3 (Ito & Murphy, 2013; Chen et al., 2022; Chen et al., 2018; Sturm, Finotello & List, 2020; R Core Team, 2020).

Statistical analysis

Multiple analytical strategies were applied for statistical evaluation. Differential expression was determined by fold-change (FC) values, whereas survival outcomes were examined using hazard ratios (HR) and log-rank tests. Associations between variables were quantified with Spearman’s or Pearson’s correlation analyses, selected according to distributional properties of the data. Statistical significance was defined as P < 0.05, derived from conventional hypothesis testing or log-rank comparisons.

Results

Screening of DENGRs in LUSC and normal lung tissues

The workflow of the study is presented in Fig. 1. A total of 159 NRGs were retrieved from the KEGG database for subsequent analysis of differential expression between LUSC and normal lung tissues. Transcriptomic profiling using the UCSC Xena database identified 5,357 DEGs, which were illustrated by a heat map (Fig. 2A) and a volcano plot (Fig. 2B). Functional enrichment analysis with KEGG and GO revealed that these DEGs were primarily associated with pathways related to the cell cycle, DNA replication, and immune response (Figs. 2C, 2D). Integration of the 159 NRGs with the 5,357 DEGs yielded 35 DENRGs through Venn diagram analysis, comprising 15 genes upregulated and 20 genes downregulated relative to normal controls, which were subsequently subjected to further investigation (Figs. 3A–3C).

Figure 1 The workflow of the present study.

Figure 2 (A) Heatmap of the 5,357 differentially expressed DEGs in LUSC. (B) Volcano plot of 5,357 differentially expressed DEGs in LUSC. (C) Enriched Gene Ontology terms and KEGG pathways associated with the 5,357 DEGs in LUSC.

Figure 3 (A) Venn diagram of the intersection of NRGs and DEGs. (B) A total of 35 NRGs among the DEGs between LUSC and normal samples. (C) The expression of 35 NRGs in LUSC and normal lung tissues, Normal, red; Tumor, blue. (D) The significant terms of KEGG analysis, accompanied by a network diagram of NRGs.

In this diagram, blue nodes denote items, red nodes represent molecules, and lines indicate the relationships between items and molecules.

Analysis of mutation and functional enrichment for 35 DENRGs

Mutation profiling and functional enrichment of 35 DENRGs in LUSC revealed distinct genetic and biological characteristics. Mutations were present in 31.71% of samples (156/492), with missense mutations representing the predominant type. Single nucleotide polymorphisms (SNPs) accounted for the majority of alterations, with C > T transitions as the most frequent single nucleotide variant (SNV) category. Among the NRGs, NLRP3, TLR4, STAT4, and PYGM exhibited the highest mutation frequencies (Figs. S1A, S1B). Functional annotation through GO and KEGG analyses delineated the biological relevance of these genes (Fig. 3D, Fig. S2). Enriched BP terms were primarily associated with apoptotic and necroptotic pathways, cellular responses to peptides, and I-kappaB kinase/NF-kappaB signaling (Fig. S2A). The encoded proteins were mainly localized within endosome membranes, membrane microdomains, membrane rafts, and the cytosolic compartment, as reflected in CC terms. MF enrichment highlighted tumor necrosis factor receptor superfamily binding, death receptor binding, ubiquitin-like protein ligase binding, and phospholipase A2 activity. Z-score clustering of GO terms indicated enrichment in necroptotic process, programmed necrotic cell death, necrotic cell death, and positive regulation of NF-kappaB transcription factor activity (Fig. S2B). KEGG pathway analysis further indicated associations with necroptosis, viral infections (Influenza A, Measles, Hepatitis, Epstein-Barr virus), NOD-like receptor signaling, TNF signaling, apoptosis, and Th1/Th2 cell differentiation (Fig. 3D). KEGG z-score clustering confirmed significant enrichment of the 35 DENRGs in pathways such as “Necroptosis” (hsa04217) and “Influenza A” (hsa05164), characterized by elevated z-scores and low P.adjust values (Fig. S2C).

Construction and validation of a prognostic nomogram incorporating NRGs in LUSC

A prognostic model for LUSC grounded in necrosis-related genes was constructed by combining LASSO regression with multivariate Cox regression analyses. This integrative strategy yielded an optimized three-gene signature, defined by CAMK2A, CHMP4C, and PYGB, with a penalty coefficient (λ) of 3 (Figs. 4A, 4B). The risk score was formulated as: Risk Score = (0.1269  × CAMK2A) + (0.2268  × PYGB) + (0.1256  × CHMP4C). Patients were subsequently categorized into high- and low-risk cohorts according to the median risk score. Figure 4C illustrates gene expression profiles, distribution of risk scores, and corresponding survival outcomes across the two subgroups. A clear positive association was observed between elevated risk scores and both higher mortality risk and shorter survival duration. Kaplan–Meier analysis revealed markedly reduced OS in the high-risk group compared with the low-risk group, with median OS times of 3.0 and 5.7 years, respectively (HR = 1.519, P = 0.0027). Time-dependent receiver operating characteristic (ROC) analysis further validated the prognostic performance of the signature, with area under the ROC curve (AUC) values of 0.588, 0.621, and 0.636 for 1-, 3-, and 5-year OS prediction, respectively (Figs. 4D–4E). To assess whether the signature provided independent prognostic information in LUSC, both univariate and multivariate Cox regression analyses were applied (Figs. 5 and 6). In the multivariate model, the risk score retained significance as an independent prognostic factor (HR = 2.76, P = 0.0012), while the pTNM stage showed only borderline significance (HR = 1.60, P = 0.01) (Figs. 5A, 5B). Based on these results, a nomogram incorporating the risk score and clinicopathological variables was established. The concordance index (C-index) of 0.627 (P < 0.05) reflected moderate predictive performance (Fig. 5C). Within the full cohort, the nomogram demonstrated reliable prediction of 1-, 3-, and 5-year OS, with estimates closely aligned to the ideal reference values (Fig. 5D).

Figure 4 Establishment of a prognostic NRGs model.

(A) LASSO coefficient profiles of three NRGs. (B) Plots of the ten-fold cross-validation error rates. (C) Distribution of risk score, survival status, and the expression of three prognostic NRGs in NRGs. D Overall survival curves for LUSC patients in the high-/low-risk group. E the ROC curve of measuring the predictive value. *P < 0.05, **P < 0.01, ***P < 0.001. Abbreviations: NRGs, Necroptosis-related gene; LUSC, lung squamous cell carcinoma; LASSO, least absolute shrinkage and selection operator; ROC, receiver operating characteristic.

Figure 5 Validation of a prognostic nomogram in LUSC.

(A) Hazard ratio and P-value of constituents involved in multivariate Cox regression and some parameters of three prognostic NRGs in LUSC. (B) Correlation between risk score and three prognostic NRGs expression levels, and distribution of risk score across clinical subgroups (pTNM stage, gender, smoking status) with survival outcome annotations (alive/dead). (C) Nomogram to predict the 1-year, 3-year, and 5-year overall survival rate of LUSC patients. (D) Calibration curve for the overall survival nomogram model in the discovery group.

Figure 6 (A–B) Univariate Cox regression analysis of three NRGs in patients with LUSC in the TCGA Database: (A) forest plot; (B) survival curve. (C) Survival analysis of three NRGs in LUSC based on the GEO database.

Univariate Cox regression analysis demonstrated the prognostic relevance of three NRGs (Figs. 6A, 6B). Elevated CAMK2A expression correlated with reduced overall survival (OS) (HR = 1.35, P = 0.029). Similarly, increased CHMP4C expression was linked to shorter OS (HR = 1.47, P = 0.005) and disease-specific survival (DSS) (HR = 1.70, P = 0.016). PYGB overexpression was associated with inferior OS (HR = 1.62, P = 0.001), progression-free survival (PFS) (HR = 1.65, P = 0.003), and DSS (HR = 2.05, P = 0.001) (Figs. 6A, 6B). Validation using GEO datasets confirmed these associations (Fig. 6C). High CAMK2A expression consistently predicted poor OS in four cohorts (GSE19188: P = 0.034; GSE41271: P = 0.010; GSE42127: P = 0.007; GSE81089: P = 0.012). CHMP4C expression correlated with inferior OS in two cohorts (GSE81089: P = 0.0084; GSE157009: P = 0.0031). PYGB was a robust indicator of poor OS in four cohorts (GSE19188: P = 0.034; GSE30219: P= 3e−04; GSE73403: P = 0.0079; GSE81089: P = 0.00096), and was further predictive of disease-free survival (DFS) in GSE30219 (P = 0.014) and recurrence-free survival (RFS) in GSE8894 (P = 0.0084). Collectively, consistent overexpression of CAMK2A, CHMP4C, and PYGB across multiple datasets indicated an unfavorable prognostic impact in LUSC across diverse survival metrics.

Mutation analysis of three NRGs in LUSC

To clarify the mutational characteristics of the three NRGs in LUSC, mutation profiles were analyzed in 18 TCGA-LUSC samples. All cases exhibited genetic alterations in CAMK2A, CHMP4C, and PYGB. The mutation spectrum comprised missense, nonsense, and multi-hit variants, with missense mutations representing the predominant type (Fig. 7A). Among the three genes, CAMK2A displayed the highest alteration frequency (50%), followed by PYGB (39%) and CHMP4C (11%). Copy number variation (CNV) assessment demonstrated extensive heterozygous deletion in CAMK2A, while CHMP4C and PYGB showed pronounced heterozygous amplification, with gene-specific CNV frequencies quantified. mRNA levels of CAMK2A and PYGB exhibited strong positive associations with CNV status, whereas CHMP4C showed only a modest correlation (Fig. 7B). Single nucleotide polymorphisms (SNPs) were the most frequent nucleotide alterations, with C>A (n = 12) and C>T (n = 6) transitions dominating the mutational landscape (Fig. 7C). Methylation profiling revealed a significant inverse relationship between methylation status and transcript levels of CHMP4C and PYGB, in contrast to CAMK2A, where no such correlation was observed. These results point to gene-specific epigenetic regulation that selectively influences the transcriptional activity of CHMP4C and PYGB (Fig. 7D).

Figure 7 Genetic and epigenetic alterations of three prognostic NRGs in LUSC: mutation profiles, CNV, and expression correlations.

(A) Mutation characteristic of three prognostic NRGs in LUSC samples, including mutation frequency and variant classification. (B) CNV frequency distribution in LUSC and correlation analysis between CNV and three prognostic NRGs mRNA expression in LUSC. (C) Variant classification distribution, mutation type prevalence, SNV class proportions, mutation burden per sample, and recurrently mutated genes. (D) Correlation between methylation status and three prognostic NRGs mRNA expression.

Analysis of mRNA and protein expression of three NRGs in LUSC

Differential expression of the three NRGs between LUSC and normal lung tissues in TCGA was verified previously (Fig. 3C). Subsequent validation using RT-qPCR and IHC was performed to evaluate their expression in clinical LUSC and normal lung samples. Quantitative RT-qPCR analysis assessed CAMK2A, CHMP4C, and PYGB mRNA levels in BEAS-2B bronchial epithelial cells and NCI-H520 LUSC cells. CAMK2A expression was significantly higher in BEAS-2B cells than in NCI-H520 cells (P < 0.01) (Fig. 8A). Conversely, CHMP4C expression was markedly increased in NCI-H520 cells, with minimal expression in BEAS-2B cells (P < 0.001) (Fig. 8B). PYGB also demonstrated elevated expression in NCI-H520 cells compared with BEAS-2B cells (P < 0.05) (Fig. 8C). Analysis of 21 paired LUSC tumor and adjacent normal tissues revealed that CHMP4C and PYGB were predominantly localized to the nucleus and cytoplasm of tumor cells, with positive immunoreactivity indicated by brown staining (Figs. 8E, 8F). In contrast, their expression was weak or undetectable in normal tissues (Figs. 8E, 8F). CAMK2A displayed the opposite pattern, with higher expression in normal tissues and reduced levels in tumor tissues (Fig. 8D). Immunohistochemical evaluation confirmed markedly increased CHMP4C and PYGB expression in LUSC compared with adjacent non-tumor tissues (P < 0.001), while CAMK2A expression was significantly enriched in normal tissues (P < 0.001) (Fig. 8G). The protein expression profiles aligned with mRNA expression data from the TCGA-LUSC cohort (Fig. 3C). Elevated CAMK2A, CHMP4C, and PYGB expression correlated with unfavorable prognosis, highlighting their value as prognostic indicators. Notably, CAMK2A was consistently higher in normal lung tissue, suggesting a potential protective role in maintaining pulmonary homeostasis that becomes attenuated during carcinogenesis. In contrast, CHMP4C and PYGB were overexpressed in LUSC and associated with poor outcomes, implying roles in tumor progression and metastatic potential. The distinct expression patterns and prognostic associations highlight the need for further mechanistic and translational investigations to clarify their contribution to LUSC pathogenesis and therapeutic sensitivity.

Figure 8 Differential expression of three NGRs at the mRNA and protein levels in LUSC.

(A) RT-qPCR showed relative mRNA expression of CAMK2A in BEAS-2B (normal bronchial epithelial cells) and NCI-H520 (LUSC cells). (B) RT-qPCR showed relative mRNA expression of CHMP4C in BEAS-2B and NCI-H520 cells. (C) RT-qPCR showed relative mRNA expression of PYGB in BEAS-2B and NCI-H520 cells. (D–F) CAMK2A, CHMP4C and PYGB protein expressions in LUSC tumor tissues and adjacent normal tissues (200× and 400× magnification). (G) Quantification of immunostains for CAMK2A, CHMP4C and PYGB by IOD analysis.

Verification of the prognostic model

The predictive model was externally validated using the GSE74777 dataset. Patients were stratified into high- and low-risk groups based on the median risk score (Figs. 9A–9C). Heatmap visualization indicated an association between risk scores and the three NRGs (Fig. 9A). ROC analysis demonstrated consistent predictive capacity for 1-, 3-, and 5-year survival across all four datasets, with AUC values exceeding 0.6 (Fig. 9D). Kaplan–Meier curves showed a trend toward reduced overall survival in the high-risk group, although statistical significance was not reached (Fig. 9E). Multivariate Cox regression further evaluated the influence of risk score alongside clinical parameters including T stage, gender, N stage, and age in LUSC patients (Fig. 9F). Collectively, the model maintained reliable performance in the GSE74777 cohort.

Figure 9 The results of various methods to verify the performance of the model based on GSE74777 dataset.

(A) Expression heat map of three NRGs. (B–C) Risk score and survival time plots. (D) Kaplan–Meier survival plot. (E) Calibration curve for the overall survival nomogram model in the discovery group. (F) Forest plots for univariate Cox regression.

The correlation between the expression levels of three NRGs and the clinical features in patients with LUSC

Analysis of TCGA-LUSC data revealed distinct prognostic patterns for the three NRGs. CAMK2A expression was significantly higher in patients older than 65 compared with those at stages I–II (P < 0.05) (Table 2). CHMP4C expression was elevated in stage III–IV patients relative to stage I–II (P < 0.05) (Table 3) and was also increased in individuals with progressive disease compared with those classified as stable disease (SD), partial response (PR), or complete response (CR) (P < 0.05) (Table 3). By contrast, PYGB expression exhibited no significant association with the clinical features of LUSC (P < 0.05) (Table 4).

Table 2 Relationship between of CAMK2A expression and clinical characteristics of patients with LUSC.

Characteristics	Total (N)	Odds Ratio (OR)	P value	
Age (>65 vs. <=65)	492	1.633 (1.036-2.696)	0.043	
Gender (Male vs. Female)	501	1.595 (0.955–2.861)	0.094	
Smoker (Yes vs. No)	489	0.871 (0.369–2.968)	0.789	
T stage (T3&T4 vs. T1&T2)	501	1.362 (0.827–2.168)	0.203	
N stage (N1&N2&N3 vs. N0)	495	0.738 (0.456–1.149)	0.196	
M stage (M1 vs. M0)	418	1.244 (0.146–4.026)	0.783	
Pathologic stage (Stage III&Stage IV vs. Stage I&Stage II)	497	1.024 (0.584–1.685)	0.928	
Primary therapy outcome (PD vs. SD&PR&CR)	361	1.202 (0.493–2.406)	0.640	

Table 3 Relationship between of CHMP4C expression and clinical characteristics of patients with LUSC.

Characteristics	Total (N)	Odds Ratio (OR)	P value	
Age (>65 vs. <=65)	492	1.114 (0.900–1.380)	0.322	
Gender (Male vs. Female)	501	0.876 (0.686–1.109)	0.278	
Smoker (Yes vs. No)	489	0.811 (0.442–1.420)	0.483	
T stage (T3&T4 vs. T1&T2)	501	1.124 (0.863–1.480)	0.393	
N stage (N1&N2&N3 vs. N0)	495	1.124 (0.904–1.406)	0.298	
M stage (M1 vs. M0)	418	1.917 (0.759–5.241)	0.192	
Pathologic stage (Stage III&Stage IV vs. Stage I&Stage II)	497	1.419 (1.067–1.914)	0.019	
Primary therapy outcome (PD vs. SD&PR&CR)	361	1.721 (1.080–2.826)	0.027	

Table 4 Relationship between of PYGB expression and clinical characteristics of patients with LUSC.

Characteristics	Total (N)	Odds Ratio (OR)	P value	
Age (>65 vs. <=65)	492	1.099 (0.892–1.356)	0.378	
Gender (Male vs. Female)	501	1.161 (0.923–1.465)	0.206	
Smoker (Yes vs. No)	489	0.890 (0.523–1.530)	0.668	
T stage (T3&T4 vs. T1&T2)	501	1.063 (0.822–1.373)	0.641	
N stage (N1&N2&N3 vs. N0)	495	1.112 (0.901–1.376)	0.323	
M stage (M1 vs. M0)	418	1.654 (0.713–3.867)	0.238	
Pathologic stage (Stage III&Stage IV vs. Stage I&Stage II)	497	1.061 (0.819–1.375)	0.652	
Primary therapy outcome (PD vs. SD&PR&CR)	361	1.382 (0.903–2.115)	0.134	

The three NRGs were associated with tumor immune infiltration in LUSC

In LUSC, the expression profiles of three NRGs display significant associations with clinical characteristics, while tumor-infiltrating lymphocytes function as independent indicators of tumor stage, grade, and lymph node involvement. Using TCGA data, the associations between the expression of these prognostic NRGs and immune infiltration were systematically examined. The tumor microenvironment, composed of malignant, stromal, and infiltrating immune cells, plays a decisive role in determining clinical outcomes, with immune infiltration recognized as an independent predictor of sentinel lymph node status and overall survival across multiple cancers. To evaluate these interactions, the ESTIMATE algorithm in R was applied to calculate immune, stromal, and ESTIMATE scores and to correlate them with the expression of CAMK2A, CHMP4C, and PYGB in LUSC. CAMK2A expression correlated positively with stromal score (R = 0.39, P = 4.7e−19), immune score (R = 0.15, P = 6.4e−4), and ESTIMATE score (R = 0.28, P = 3.2e−10), suggesting its involvement in promoting stromal and immune activity. In contrast, CHMP4C expression exhibited negative correlations with all three metrics, including immune score (R = −0.25, P = 1.5e−8), stromal score (R = −0.24, P = 8.0e−8), and ESTIMATE score (R = −0.24, P = 1.2e−5), reflecting an opposing relationship with stromal and immune components. PYGB displayed weaker associations, with a minor negative correlation with the immune score (R = −0.15, P = 1.2e−3) and no statistically significant associations with stromal or ESTIMATE scores (P = 0.82 and P = 0.09, respectively).

Given the strong association of the three NRGs with immune infiltration, their immunological relevance in LUSC was further examined using TCGA data and single-sample Gene Set Enrichment Analysis (ssGSEA) analysis (Fig. 10B). CAMK2A displayed an immunostimulatory pattern, showing positive correlations with NK cells (R = 0.470, P < 0.001), macrophages (R = 0.344, P < 0.001), Th1 cells (R = 0.327, P < 0.001), Tem cells (R = 0.295, P < 0.001), along with fifteen additional immune cell types. In contrast, CHMP4C exhibited an immunosuppressive profile, with negative correlations involving Tregs (R = −0.283, P < 0.001), B cells (R = −0.255, P < 0.001), and pDCs (R = −0.255, P < 0.001). PYGB showed a dual regulatory influence, positively correlated with NK cells (R = 0.155, P < 0.001), but negatively correlated with helper T cells (R = −0.249, P < 0.001), cytotoxic cells (R = −0.223, P < 0.001), and CD8+ T cells (R = −0.234, P < 0.001). Collectively, the analysis revealed distinct immunomodulatory patterns, with CAMK2A associated with pro-inflammatory responses, PYGB reflecting mixed regulatory activity, and CHMP4C predominantly linked to immunosuppression. The robustness of these results was further confirmed through CIBERSORT, which validated the immune landscape characterized by ssGSEA and demonstrated concordance across analytical methods (Fig. 10C). Continued investigation into the correlation between these NRGs and immune infiltration in LUSC is warranted to consolidate their functional associations with tumor-infiltrating immune cells.

Figure 10 (A) The correlation between three prognostic NRGs and tumor microenvironment scores, as determined by the ESTIMATE algorithm, highlighting their association with immune, stromal, and ESTIMATE scores in LUSC. (B) The relationship between the expression levels of these three prognostic NRGs and immune infiltration in LUSC, analyzed using the ssGSEA algorithm. (C) The correlation between the expression levels of the three prognostic NRGs and immune infiltration in LUSC, as assessed by the CIBERSORT algorithm.

*P < 0.05, **P < 0.01, ***P < 0.001. Abbreviations: NRGs, necroptosis-related gene; LUSC, lung squamous cell carcinoma; TILs, tumor-infiltrating lymphocytes.

Immune-related genes TMB and MSI analysis of the three NRGs

To delineate the relationship between NRGs and immune infiltration in LUSC, mRNA expression patterns of CAMK2A, CHMP4C, and PYGB were systematically correlated with immune-related gene sets, including chemokines, chemokine receptors, major histocompatibility complex (MHC) molecules, immunoinhibitors, and immunostimulators, across 32 TCGA cancer types (Fig. 11A). CAMK2A displayed broad positive correlations, with distinct clusters of strong associations in several cancers, whereas CHMP4C and PYGB exhibited heterogeneous profiles characterized by both positive and negative associations. CAMK2A was most closely linked with chemokine receptors such as CXCR4 and CCR5, while CHMP4C and PYGB were more strongly associated with MHC class I/II components and immunoinhibitory molecules, suggesting context-dependent functions in immune cell trafficking and antigen presentation.

Figure 11 (A) The relationship between the expression levels of these three prognostic NRGs and Immune-related genes in pan-cancers; (B) The relationship between the three prognostic NRGs expression levels and immune checkpoints in LUSC. (C) The correlation of three prognostic NRGs with TMB and MSI in LUSC.

*P < 0.05, **P < 0.01, ***P < 0.001. Abbreviations: NRGs, Necroptosis-related gene; LUSC, lung squamous cell carcinoma.

Subsequent evaluation of the three NRGs in relation to immune checkpoint genes in LUSC (Fig. 11B) revealed that CAMK2A expression correlated positively with SIGLEC15, CTLA4, HAVCR2, IGSF8, PDCD1, PDCD1LG2, and TIGIT. CHMP4C was positively associated with IGSF8 but negatively correlated with CD274, CTLA4, HAVCR2, ITPRIP1, LAG3, PDCD1, PDCD1LG2, and TIGIT. PYGB demonstrated positive correlation with IGSF8 and negative correlation with CTLA4 and TIGIT. Collectively, these patterns indicate a regulatory involvement of CAMK2A, CHMP4C, and PYGB in immune checkpoint pathways in LUSC. Given the relevance of TMB and MSI as biomarkers for immunotherapy response, their associations with NRGs were further assessed. CAMK2A exhibited a significant correlation with TMB, while CHMP4C and PYGB showed no such association (Fig. 11C). None of the three NRGs demonstrated a correlation with MSI.

Discussion

Necroptosis, a recently characterized form of programmed cell death, exhibits features of both apoptosis and necrosis and exerts broad influence on tumorigenesis, proliferation, invasion, and metastasis, thereby shaping cancer prognosis (Zhang et al., 2022). Its role in oncology is inherently dual. Activation of necroptosis can trigger adaptive immune responses that enhance antitumor immunity and suppress tumor progression (Ye, Chen & Xu, 2023), yet it has also been linked to enhanced invasiveness, metastatic potential, and unfavorable survival outcomes across multiple cancer types. Resistance to apoptosis-driven therapies, a frequent hallmark of multidrug-resistant tumors, further diminishes the efficacy of conventional treatment modalities (Chen et al., 2019). Necroptosis provides an apoptosis-mimicking mechanism that offers potential to overcome apoptotic resistance and selectively eliminate malignant cells. Regulators of necroptotic signaling, including RIPK1, RIPK3, and MLKL, have demonstrated prognostic relevance in diverse tumor entities (Yuan, Amin & Ofengeim, 2019; Martens et al., 2021; Zang et al., 2022). This pathway is particularly pertinent in LUSC, where evasion of canonical apoptotic programs is common, thereby highlighting the necessity of exploring alternative therapeutic avenues (Krysko et al., 2017; Najafov, Chen & Yuan, 2017). Notably, reduced expression of RIPK3 and MLKL has been documented in LUSC and correlates with inferior prognosis (Park et al., 2020; Lim et al., 2021), supporting necroptosis as a promising therapeutic target. However, the clinical significance of NRGs in LUSC remains insufficiently defined, necessitating deeper investigation.

Differential expression analysis was performed to compare NRGs between LUSC and normal lung tissues. From 159 NRGs obtained from the KEGG database and 5,357 DEGs retrieved from the UCSC Xena database, intersection analysis identified 35 DENRGs, comprising 15 upregulated and 20 downregulated genes. A prognostic model for LUSC was then established by combining LASSO and multivariate Cox regression analyses, which highlighted CAMK2A, CHMP4C, and PYGB as key components of a necroptosis-associated risk signature. Elevated expression of these three genes correlated with unfavorable prognosis in LUSC, supporting their potential as prognostic indicators. Notably, CAMK2A exhibited higher expression in normal lung tissues, whereas CHMP4C and PYGB were markedly overexpressed in LUSC, with their dysregulation associated with poorer clinical outcomes and a potential role in tumor progression and metastasis.

CAMK2A functions as a multifunctional enzyme engaged in diverse signaling pathways that regulate cellular proliferation, differentiation, and survival (Küry et al., 2017). Aberrant expression and activity of this kinase have been linked to both physiological regulation and pathological states, including oncogenesis. Evidence indicates that CAMK2A promotes tumor-initiating capacity in lung adenocarcinoma by enhancing SOX2 expression through EZH2 phosphorylation, highlighting its complex involvement in tumor biology (Wang et al., 2020). Consistent with this evidence, the present analysis indicates an oncogenic role for CAMK2A. Notably, higher CAMK2A expression was detected in normal lung tissues compared with cancerous tissues, suggesting a protective function in maintaining normal lung homeostasis that was diminished during malignant transformation. Expression patterns further demonstrated tissue specificity, exemplified by its correlation with NF2 in meningiomas and normal nervous system tissues, whereas no such relationship was evident in tumors of non-neural origin (Lei, Cai & Yan, 2024). Such tissue-dependent regulation suggests context-specific functions of CAMK2A, which may account for its differential expression between normal and malignant lung tissues.

CHMP4C, a key element of the ESCRT-III complex, is frequently upregulated in multiple malignancies. In pancreatic cancer, its overexpression accelerates tumor progression by inhibiting necroptosis through the RIPK1/RIPK3/MLKL signaling cascade (Yu et al., 2025). Within lung cancer, CHMP4C disruption has been reported to increase radiosensitivity, indicating a role in mediating resistance to irradiation and highlighting its therapeutic relevance (Li et al., 2015). Beyond this, CHMP4C participates in intracellular trafficking and signaling regulation, particularly within the EGFR pathway, thereby contributing to oncogenic progression and unfavorable prognosis (Liu et al., 2023). Consistent with these reports, the present study identified CHMP4C overexpression in LUSC, with elevated levels correlating with adverse clinical outcomes and suggesting a contribution to both tumor progression and metastatic potential.

PYGB has been implicated in the growth and progression of multiple malignancies. In gastric cancer, PYGB promotes proliferation, invasion, and migration through regulation of the Wnt-β-catenin signaling pathway (Xia, Zhang & Liu, 2020). Li et al. (2020) identified an association between PYGB expression and smoking, linking it to LUSC progression. In prostate cancer, PYGB activation via NF-κB/Nrf2 signaling contributes to enhanced proliferation, migration, and aggressiveness (Wang et al., 2018). Consistent with these observations, elevated PYGB expression in LUSC correlated with poor prognosis.

Prognostic relevance was further examined through univariate and multivariate Cox regression analyses of the three NRGs signature, while RT-qPCR and IHC confirmed differential expression between LUSC and normal samples. A nomogram integrating the risk score with clinicopathological features achieved a C-index of 0.627 (P < 0.05), reflecting moderate predictive power. This model accurately estimated 1-, 3-, and 5-year OS rates with close alignment to actual outcomes, and its prognostic performance was validated using an external dataset.

The interplay between necroptosis and immune regulation has attracted increasing attention in recent years. As a programmed form of necrotic cell death, necroptosis exerts broad effects on both innate and adaptive immunity, particularly in cancer and infectious diseases (Meier et al., 2024). Within oncology, necroptosis may enhance antitumor immune activity and constitutes a therapeutic strategy against cancer cells resistant to apoptosis (Zhang et al., 2024). In LUSC, the tumor immune microenvironment critically influences tumor initiation, progression, and therapeutic response. Evidence indicates a strong association between the immune landscape of LUSC and the degree of immune cell infiltration (Lahiri et al., 2023). Variations in the composition and functional characteristics of infiltrating cells directly affect prognosis and the effectiveness of immunotherapies.

To further explore these mechanisms, the association between five prognostic NRGs and immune cell infiltration in LUSC was assessed. Distinct correlations emerged between gene expression and stromal, immune, and estimate scores, highlighting complex interactions between tumor microenvironmental components and gene activity. CAMK2A displayed a moderate positive correlation with all three scores, suggesting involvement in sustaining stromal and immune elements within the microenvironment. This association implies that CAMK2A may regulate signaling cascades governing immune cell recruitment or stromal cell proliferation, thereby shaping both tumor dynamics and host immune responses. Detailed investigation of the molecular mechanisms by which CAMK2A influences stromal and immune compartments may reveal novel therapeutic targets for modulating the tumor microenvironment. In contrast, the consistent negative correlations of CHMP4C with immune, stromal, and ESTIMATE scores indicate a potential inhibitory role in stromal development and immune cell recruitment or activation. CHMP4C may operate through signaling pathways that suppress immune responses or limit stromal cell proliferation, thereby fostering an immunosuppressive microenvironment favorable to tumor persistence. Clarifying its mechanistic contribution could support the design of interventions aimed at mitigating immunosuppression and strengthening anti-tumor immunity. PYGB, by comparison, showed only weak correlations, including a marginal negative association with immune score, suggesting a more restricted influence on the tumor microenvironment than CAMK2A or CHMP4C. Nonetheless, even modest effects may alter immune dynamics, possibly through metabolic pathways that indirectly affect immune cell viability or activity. Although no significant relationships with stromal or ESTIMATE scores were observed, further analysis of subtle regulatory roles remains warranted. Future research should incorporate functional experiments, such as gene knockdown or overexpression, to delineate the specific contributions of these genes within the tumor microenvironment and assess their potential as biomarkers or therapeutic targets. Examination of their co-expression with established tumor-related genes may also provide broader insights into the regulatory networks shaping tumor-immune and stromal interactions.

The distinct immunomodulatory patterns of CAMK2A, CHMP4C, and PYGB indicate their potential as regulators within the complex interface between immune activity and cellular processes, warranting deeper exploration of their molecular mechanisms. The consistent positive associations of CAMK2A with diverse immune cell populations suggest an immunostimulatory function, likely involving activation and recruitment of NK cells, macrophages, and Th1 cells. Clarifying the signaling pathways through which CAMK2A exerts these effects, including possible interactions with cytokines or chemokines, could reveal how its activity promotes immune cell migration and functional responses. Elucidating upstream regulators of CAMK2A expression would also improve understanding of its modulation under physiological and pathological contexts. In contrast, the negative correlations of CHMP4C with Tregs, B cells, and pDCs suggest a suppressive influence on immune responses, potentially contributing to tolerance or limiting excessive immune activation. Examination of CHMP4C’s impact on cytokine secretion, antigen presentation, and related immune functions may clarify its role in dampening immunity, while analysis of its regulation in autoimmune or chronic infectious states could uncover therapeutic avenues. PYGB demonstrated a more complex profile, with positive association to NK cells but negative correlations with Th, cytotoxic, and CD8+ T cells, implying a dual role in maintaining immune balance. By enhancing NK-mediated immunity while restraining T cell-driven responses, PYGB may act as a metabolic checkpoint. Given its role in glycogen metabolism, PYGB could influence the energetic state of immune cells, thereby shaping their activation thresholds and effector functions. Investigation of this metabolic–immune interplay would refine the understanding of PYGB’s contribution to immune regulation.

Correlations between the three NRGs and immune checkpoint genes emphasized the interplay between necroptosis-related pathways and immune regulation. CAMK2A exhibited strong positive associations with multiple checkpoints, including SIGLEC15, CTLA4, HAVCR2, and PDCD1, suggesting a role in reinforcing immune suppressive mechanisms and attenuating T-cell activity. Such associations position CAMK2A as a regulator within the checkpoint network, potentially influencing tumor immune evasion. Mechanistic clarification regarding whether this regulation occurs through transcriptional control or epigenetic modification remains an important direction for investigation. CHMP4C demonstrated a more complex correlation profile. Its positive association with IGSF8 implies a contribution to immune cell adhesion and migration, while negative correlations with CD274 (PD-L1), CTLA4, and LAG3 suggest a capacity to diminish checkpoint activity. This dual pattern highlights CHMP4C as a potential therapeutic target for enhancing anti-tumor immunity by limiting inhibitory signaling within the microenvironment. Further exploration of CHMP4C in specific immune cell subsets and its impact on tumor-immune dynamics is warranted. PYGB presents a distinct regulatory profile, with positive correlation to IGSF8 and negative correlation to CTLA4 and TIGIT. The association with IGSF8 may support immune cell recruitment and activation at inflammatory or tumor sites, whereas negative associations with CTLA4 and TIGIT imply a counter-regulatory effect on checkpoint pathways, potentially strengthening T-cell-mediated anti-tumor responses.

Although validation was conducted using multiple databases, RT-qPCR, and IHC, certain limitations persist. A considerable part of the analyses depended on publicly available datasets, raising the possibility of case selection bias that might affect the conclusions. To strengthen the reliability of the results, large-scale prospective investigations and additional in vitro and in vivo studies remain necessary.

In conclusion, an NRG signature comprising CAMK2A, CHMP4C, and PYGB is established as a prognostic predictor in LUSC. Expression patterns of these prognostic NRGs are corroborated by RT-qPCR and immunohistochemistry. Moreover, the analyses suggest that these NRGs may influence LUSC pathogenesis through modulation of tumor immune infiltration and immune checkpoint expression. Further mechanistic studies and clinical trials are required to refine understanding and validate the translational potential of this signature.

Supplemental Information

Supplemental Information 1 Variant Distribution, Classification, and Top Mutated Genes A. Frequency of genetic alterations in 492 samples B. Variant classification and type details and summary of top 10 mutated genes with their mutation rates

Supplemental Information 2 GO and KEGG analysis of differentially expressed NRGs in LUSC . A The enrichment of genes across biological processes, cellular components, and molecular functions, accompanied by a network diagram of NRGs. In this diagram, blue nodes denote items, red

Supplemental Information 3 159 NRGs from KEGG in LUSC

Supplemental Information 4 35 NRGs among the DEGs between LUSC and normal samples

Supplemental Information 5 Clinical characteristics of patients with LUSC

Supplemental Information 6 IHC DATA

Supplemental Information 7 Q-PCR DATA

Supplemental Information 8 TCGA RAW DATA

Supplemental Information 9 raw data

Additional Information and Declarations

Competing Interests

Author Contributions

Human Ethics

Data Availability

The authors declare there are no competing interests.

Kai Sun conceived and designed the experiments, performed the experiments, analyzed the data, prepared figures and/or tables, authored or reviewed drafts of the article, and approved the final draft.

Ke-Run Wang performed the experiments, prepared figures and/or tables, authored or reviewed drafts of the article, and approved the final draft.

Song Wen performed the experiments, authored or reviewed drafts of the article, and approved the final draft.

Juan-Juan Hong performed the experiments, authored or reviewed drafts of the article, and approved the final draft.

Yu-Lang Fei analyzed the data, authored or reviewed drafts of the article, and approved the final draft.

Qing-Hua Pan conceived and designed the experiments, analyzed the data, authored or reviewed drafts of the article, and approved the final draft.

Fang-Fang Xie conceived and designed the experiments, prepared figures and/or tables, and approved the final draft.

The following information was supplied relating to ethical approvals (i.e., approving body and any reference numbers):

The Medical Ethics Committee of Liuzhou People’s Hospital (Reference No. KY2021-025-02 authorized this study.

The following information was supplied regarding data availability:

The raw data is available in the Supplemental Files.

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
