# Peer review of "Identification and experimental verification of necroptosis-related prognostic gene signature and characterization of tumor immune infiltration in lung squamous cell carcinoma"

_PeerJ, doi:10.7717/peerj.20260_

## Round 0.1 · original submission · Major Revisions

The manuscript requires major revisions as recommended by the reviewers. I urge the authors to skeptically revise the manuscript and provide a point by point explanations to the reviewers concern.

Reviewer 1 ·

Basic reporting

This study investigates the role of necroptosis-related genes (NRGs) in lung squamous cell carcinoma (LUSC). Three NRGs—CAMK2A, CHMP4C, and PYGB—were identified as significant prognostic indicators and used to build a risk model that effectively stratifies patients into high- and low-risk groups with distinct survival outcomes. Additionally, these genes were linked to immune infiltration, checkpoint, tumor mutation burden, and were aberrantly expressed in LUSC, highlighting their potential as therapeutic targets. This manuscript presents a potentially impactful study on necroptosis-related gene signatures in LUSC. While the findings are valuable, the manuscript would benefit from several revisions to improve clarity, scientific rigor, and overall presentation.
Major Comments:
1 In Figure 1, it is unclear what defines Group G1. The manuscript should explicitly describe how cohort G1 and the normal group are defined (e.g., total number, patient demographics, clinical criteria, etc).

2 Figure 3 & Figure 4: There is a lack of clear separation of results by panel in the text. Please reference each panel explicitly (e.g., Fig. 3A, 3B...) and ensure the results align clearly with the corresponding figure.

3 Figure 7:The text mentions comparison to an "ideal model," but the criteria for this comparison are not described. Please clarify.

4 Figure 8: Details regarding the external datasets GSE74777 and GSE53624 are insufficient. Please include information on sample size, study origin, and reasons for their selection as validation datasets.

5 Figure 9:
o The scale bar is missing from the immunohistochemistry images.
o CAMK2A expression appears elevated in LUSC tissue based on the image, which contradicts the claim in the text. Please clarify and ensure consistency between the image interpretation and discussion.
o Since all three NRGs are discussed as pro-tumorigenic, it would help to clarify their expression profiles with this interpretation in both the results and discussion sections.

6 Figure 10:
o Please briefly explain what CIBERSORT does.
o The biological implications of NRG’s association with immune infiltration, immune checkpoint etc. are not well discussed. The discussion section does not clearly articulate this point and would benefit from clearer explanation.

Minor comments:
• The manuscript would benefit from thorough English language and grammar editing throughout to improve readability and precision.
• Line 581: "Bladder" appears to be a typographical error; it likely should be "lung"?
• Line 219, Line 254 and Line 221: There are typos and dataset labeling or referencing errors—please double-check
• Line 262: Clarify whether the reported p-values apply only to PYGB?
• PDF figures are blurred in the main document, although they appear clear when viewed individually. Please ensure all figures are high-resolution in the final submission.

Experimental design

NA

Validity of the findings

NA

Additional comments

NA

Reviewer 2 ·

Basic reporting

No comment

Experimental design

While the authors present mutation frequency data for the necroptosis-related genes, the study would be enhanced by including correlation analysis between mutation status and gene expression levels, which could provide valuable mechanistic insights into whether genetic alterations drive the observed transcriptomic changes and contribute to the prognostic significance of these genes.

Validity of the findings

No comment

·

Basic reporting

The manuscript entitled “Identification and experimental verification of necroptosis-related prognostic gene signature and characterization of tumor microenvironment infiltration in lung squamous cell carcinoma” addresses an important topic with significant scientific interest. In this study, the authors report the prognostic significance of a necroptosis-related gene signature comprising three genes CAMK2A, CHMP4C and PYGB in LUSC using various bioinformatics tools and IHC analysis in patient tissues. The findings suggest that these genes are involved in immune microenvironment modulation and may serve as potential prognostic biomarkers in LUSC. Overall, the manuscript is thoughtfully designed. However, there are some overall shortcomings and the data are not sufficient to support the conclusions. Therefore, it is recommended that the manuscript be accepted after major revisions for publication. The following issues should be addressed:
Major Comments
• The rationale for focusing on necroptosis in LUSC is not adequately explained. The authors should provide a clearer justification for why necroptosis-related genes were chosen as the focus, supported by relevant background literature.
• In the abstract, abbreviations such as LUSC and NRGs should only be defined upon first use and not repeated in parentheses in subsequent mentions.
• The figures throughout the manuscript are of low resolution, and the font sizes are too small to read clearly. Please revise all figures for better clarity and readability.
• While Figure 1C presents multiple enriched pathways, it is unclear why necroptosis was selected for further analysis in Figure 2 and beyond. The authors should clarify why necroptosis was prioritized over other enriched pathways.
• Figures 2, 3, and 4 (except Figure 4A) do not add significant impact to the manuscript and may be moved to the supplementary section. Figure 4A, which highlights necroptosis as a top pathway, can remain in the main text.
• In Figure 7 and the corresponding legend, the authors mention Cox regression results for CESC (cervical squamous cell carcinoma), which appears irrelevant to this LUSC-focused study. Please clarify.
• The IHC results are presented abruptly. Specifically, the text mentions that CAMK2A expression is higher in normal tissues, yet the representative image does not reflect this. Please reconcile the discrepancy between the image and quantification. Also, references to CESC and cervical cancer are confusing and should be either explained or removed.
• Introduction: Please cite a recent cancer statistics review (e.g., Cancer Statistics 2024) to update and strengthen the relevance of your background section.
• The authors state that "many potential biomarkers exist for LUSC prognosis, but none are clinically applicable." Please name a few such biomarkers with appropriate references to support this claim and improve the logical flow for readers.
• In line 221, "patients with LUSA" appears to be a typographical error. Please clarify or correct.
• In the results section, the authors should state the purpose of each analysis, what they aim to investigate, and how it logically connects to the next set of experiments. Improving the narrative structure will greatly enhance readability.
• To strengthen the experimental evidence, the expression levels of CAMK2A, CHMP4C, and PYGB in LUSC and normal tissues or cell lines should be validated using qPCR.

Experimental design

no comment

Validity of the findings

no comment

Additional comments

no comment

---

## Round 0.2 · Minor Revisions

We appreciate the author's input in acknowledging the reviewers' comments.
However, the reviewers have raised minor concerns that need attention.

**Language Note:** The review process has identified that the English language must be improved. PeerJ can provide language editing services - please contact us at [email protected] for pricing (be sure to provide your manuscript number and title). Alternatively, you should make your own arrangements to improve the language quality and provide details in your response letter. – PeerJ Staff

Reviewer 1 ·

Basic reporting

The authors have addressed most of the earlier comments; however, the manuscript still requires revision. Several line numbers cited in the rebuttal are incorrect, which made it inconvenient to locate the revised sections. Please verify all cross-references and ensure that changes described in the rebuttal are reflected in the manuscript. A careful English-language edit is also recommended before resubmission.
Minor:
Regarding #2: Fig. S2 is not described in the main text. Although it is explained in the rebuttal, the description should also be included within the manuscript for completeness.

Regarding #4: Fig. 8 -the phrasing provided in the rebuttal does not exactly appear in the main text. Please ensure consistency between the manuscript and the rebuttal.

Regarding #6: The use of CIBERSORT is mentioned, but no explanation is provided in the main text. A brief description of the method should be included to improve clarity for readers unfamiliar with the approach.

Typographical error: The term “bladder/LUSA” still appears in the manuscript and should be corrected.

Experimental design

NA

Validity of the findings

NA

Additional comments

NA

Reviewer 2 ·

Basic reporting

No Comment

Experimental design

The authors have addressed the concerns from the previous review. I appreciate their effort and recommend publication of the manuscript in its current form.

Validity of the findings

No Comment

---

## Round 0.3 · accepted · Accept

The authors have shown their efforts in accomodating all the comments.